# POINT CLOUD DATASET DISTILLATION

## ABSTRACT

This study introduces dataset distillation (DD) tailored for 3D data, particularly point clouds. DD aims to substitute large-scale real datasets with a small set of synthetic samples while preserving model performance. Existing methods mainly focus on structured data such as images. However, adapting DD for unstructured point clouds poses challenges due to their diverse orientations and resolutions in 3D space. To address these challenges, we theoretically demonstrate the importance of matching rotation-invariant features between real and synthetic data for 3D distillation. We further propose a plug-and-play point cloud rotator to align the point cloud to a canonical orientation, facilitating the learning of rotation-invariant features by all point cloud models. Furthermore, instead of optimizing fixed-size synthetic data directly, we devise a point-wise generator to produce point clouds at various resolutions based on the sampled noise amount. Compared to conventional DD methods, the proposed approach, termed DD3D, enables efficient training on low-resolution point clouds while generating high-resolution data for evaluation, thereby significantly reducing memory requirements and enhancing model scalability. Extensive experiments validate the effectiveness of DD3D in shape classification and part segmentation tasks across diverse scenarios, such as cross-architecture and cross-resolution settings.

## 1 INTRODUCTION

Dataset distillation (DD) (Wang et al., 2018) aims to distill the knowledge of a large-scale dataset into a few synthetic samples, where the models trained on the real and synthetic data will have comparable performance. By doing so, DD significantly reduces the computational cost of training neural networks from scratch. Due to its remarkable efficiency and effectiveness, DD has been used in a variety of domains, such as image (Zhao et al., 2021; Zhao & Bilen, 2023; Cazenavette et al., 2022), video (Wang et al., 2024), text (Maekawa et al., 2023) etc. Despite great progress, existing DD methods only succeed on structured 1D and 2D data, while the distillation of unstructured 3D data, *e.g.*, point cloud, is still under-explored.

Point cloud data exists in large quantities in various fields. For example, MVPNet (Yu et al., 2023) scans more than 87K point clouds from real-world videos for machine vision, Objaverse-XL (Deitke et al., 2023) provides more than 10M high-quality 3D assets, and Qu et al. (2022) constructs a 100M dataset for high-energy physics, where particles are modeled as point clouds. Training on these datasets from scratch is time-consuming and resource-intensive, requiring more efficient approaches. However, several reasons prevent existing DD frameworks from generalizing to 3D point clouds.

First, point clouds with different orientations represent the same semantic information, *e.g.*, shapes. However, existing DD methods do not take the symmetry of data into account, which cannot handle the randomly rotated data and result in sub-optimal performance. As shown in Figure 1a, directly applying DD to the point clouds with different orientations cannot obtain meaningful synthetic data. Second, point clouds have flexible resolutions, *i.e.*, the number of points, depending on specific models and applications. Generally, a larger resolution encodes more fine-grain information but also increases the computational costs (Huang et al., 2024; Qiu et al., 2021). However, existing DD methods initialize the synthetic data as a fixed-size tensor, which cannot be customized for different point cloud models. Moreover, the memory budget for fixed-size tensors will increase rapidly when dealing with dense-resolution scenes, *e.g.*, segmentation (Chang et al., 2015; Ren et al., 2022).

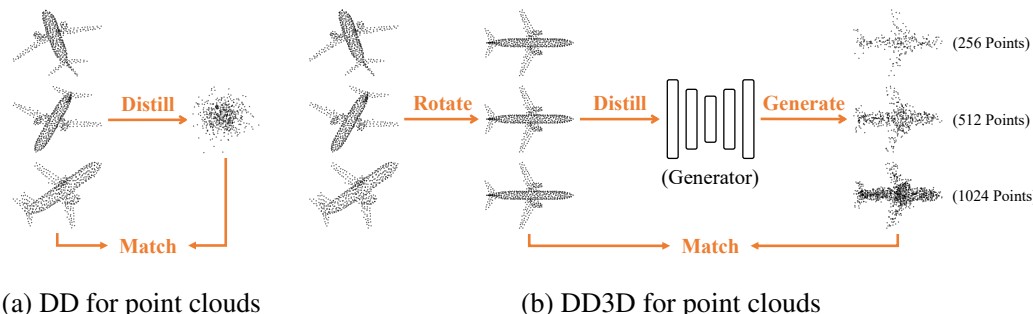

(a) DD for point clouds      (b) DD3D for point clouds

Figure 1: Differences between vanilla DD and DD3D when distilling 3D point clouds.

Once the weaknesses of existing methods are identified, it is natural to ask: *How can we build a distillation framework that overcomes the orientation and resolution issues of 3D point clouds?* To answer this question, we first theoretically prove that random rotations weaken the principle components of real data, thereby degenerating the distillation performance. Based on this discovery, we propose **DD3D**, the first DD framework for 3D point clouds, illustrated in Figure 1b. Specifically, DD3D first uses a rotator to convert the point cloud into a canonical orientation by learning a rotation-equivariant projection matrix to offset random rotation. Then, the knowledge of rotation-invariant data is distilled into a point-wise generator to predict the point coordinates from noise, where the resolution is based on the number of sampled noises. Finally, the rotator and generator are jointly optimized by minimizing the gradient differences between the real and synthetic data.

The contributions are summarized as follows. (1) We propose the first 3D distillation framework, DD3D, which can eliminate the influence of random rotations and synthesize point clouds at arbitrary resolutions. (2) We theoretically prove that matching the rotation-invariant features can preserve the principal components of real data and prevent data degeneration. (3) DD3D can be trained with low-resolution point clouds and generates high-resolution data for evaluation, significantly reducing memory usage and enhancing model scalability. (4) Extensive experiments on shape classification and part segmentation tasks validate the effectiveness of DD3D over baselines.

## 2    RELATED WORK

**Dataset Distillation.** Research on DD can be roughly divided into two directions. The first is to explore advanced matching objectives to improve the distillation performance. For example, performance matching (Wang et al., 2018), gradient matching (Zhao et al., 2021; Zhao & Bilen, 2021), distribution matching (Zhao & Bilen, 2023; Wang et al., 2022), trajectory matching (Cazenavette et al., 2022; Guo et al., 2024; Du et al., 2023) and feature regression (Zhou et al., 2022; Loo et al., 2022; Nguyen et al., 2021). On the other hand, some methods innovate efficient data parameterizations to avoid directly optimizing the synthetic data. For example, neural networks (Liu et al., 2022), spectral representation (Shin et al., 2023), linear transformation (Deng & Russakovsky, 2022), and up-sampling (Kim et al., 2022). Among them, a special parameterization technique is to distill the knowledge into a generative model (Zhao & Bilen, 2022; Wang et al., 2023; Zhang et al., 2023; Cazenavette et al., 2023; Zhang et al., 2024), which can generate diverse synthetic data with unlimited samples. Although valid, these methods rely on the prior knowledge of generative models pre-trained on large-scale datasets, which is not feasible for point clouds. A recent work[1] also applies GM to point cloud data. However, neither of them considers the orientation and resolution issues. For a more comprehensive introduction to DD, please refer to the recent surveys (Yu et al., 2024; Lei & Tao, 2024; Geng et al., 2023; Sachdeva & McAuley, 2023).

**Point Cloud Analysis.** Deep learning on point clouds plays a vital role in 3D data analysis (Guo et al., 2021b). Traditional methods can be classified into three categories: Point-based methods, *e.g.*, PointNet (Qi et al., 2017a) and PointNet++ (Qi et al., 2017b), convolution-based methods, *e.g.*, PointCNN (Li et al., 2018) and PointConv (Wu et al., 2019), and relation-based methods, *e.g.*, DGCNN (Wang et al., 2019) and Point Transformer (Guo et al., 2021a). However, these methods

---

[1] https://github.com/kghandour/dd3d

are rotation-sensitive and cannot handle point clouds with different orientations. Some advanced methods are designed to learn rotation-equivariant or invariant features, such as vector neuron (Deng et al., 2021), spherical harmonic (Poulenard et al., 2019), tensor field (Thomas et al., 2018), and graph features (Kim et al., 2020; Zhao et al., 2019). However, these methods introduce additional operators and cannot preserve the original geometric information, *i.e.*, coordinates. Another way is to project point clouds into the same orientation. For example, principal component analysis (PCA) leverages the eigenvectors of the covariance matrix to transform point clouds into the direction with maximum variance. But this approach suffers from the sign-ambiguity issue (Xiao et al., 2020; Yu et al., 2020; Li et al., 2021).

## 3 BACKGROUND

### 3.1 PRELIMINARY

**Task Formulation.** Suppose that $\mathcal{T} = \{(\mathcal{C}_i, y_i)\}_{i=1}^{|\mathcal{T}|}$ is a large-scale training dataset, where $\mathcal{C}_i$ is a point cloud with label $y_i$ for the shape classification task. Each point cloud has $n$ points, represented as $\mathcal{C} = \{P, V\}$, where $P \in \mathbb{R}^{n \times 3}$ represents the 3D coordinates of points and $V \in \mathbb{R}^{n \times v}$ indicates the part to which the point belongs in segmentation task and $v$ is the number of parts. The goal of DD3D is to synthesize a much smaller point cloud dataset $\mathcal{S} = \{(\mathcal{C}_j, y_j)\}_{j=1}^{|\mathcal{S}|}$, where $|\mathcal{S}| \ll |\mathcal{T}|$, such that a classification or segmentation model $f_\theta$ trained on $\mathcal{T}$ and $\mathcal{S}$ will have comparable performance. Other tasks, such as detection, are left for future studies.

**Dataset Distillation.** In order to effectively optimize the synthetic data, existing DD methods adopt a bi-level optimization paradigm, which can be formulated as:

$$\min_{\mathcal{S}} \mathcal{L}_{DD}\left(f_{\theta*}(\mathcal{S}), f_{\theta*}(\mathcal{T})\right) \quad \text{s.t.} \quad \theta* = \arg\min_{\theta} \mathcal{L}_{cls}(f_\theta(\mathcal{S}), Y^{\mathcal{S}}), \quad (1)$$

where the inner loop updates the model $f_\theta$ on the synthetic data, and the outer loop optimizes the synthetic data. In particular, $\mathcal{L}_{DD}$ is a metric that measures the distance between real and synthetic data. For example, gradient matching (Zhao et al., 2021) minimizes the gradient differences.

### 3.2 DATASET DISTILLATION WITH ROTATIONS

Before detailing the proposed method, we first give a general analysis of how rotations affect the performance of DD. Let $X_{\mathcal{S}} \in \mathbb{R}^{|\mathcal{S}| \times d}$, $X_{\mathcal{T}} \in \mathbb{R}^{|\mathcal{T}| \times d}$ denote the representations learned by $f_\theta$ on the synthetic data and real training data, respectively, and $d$ is the hidden dimension.

**Theorem 1.** *Assume the classifier is a linear layer $W$ and $\mathcal{L}_{cls}$ can be simplified to the mean-squared error $\|XW - Y\|_F^2$. The objective of gradient matching is equal to variance preserving:*

$$\min_{\mathcal{S}} \mathcal{L}_{GM} = \min_{\mathcal{S}} \mathcal{D}\left(\nabla_W \mathcal{L}_{cls}^{\mathcal{S}}, \nabla_W \mathcal{L}_{cls}^{\mathcal{T}}\right) \quad \Rightarrow \quad \min_{\mathcal{S}} \left\|X_{\mathcal{S}}^\top X_{\mathcal{S}} - X_{\mathcal{T}}^\top X_{\mathcal{T}}\right\|_F^2, \quad (2)$$

*where $\mathcal{D}$ is a distance metric and $\nabla_W$ is the gradient with respect to $W$.*

Theorem 1 reveals that synthetic data preserves the variance information of real data. We then analyze how random rotations affect the variance of real data. Without loss of generality, we assume that $f_\theta$ is rotation-equivariant, *i.e.*, $f_\theta(PR) = f_\theta(P)R$, where $R \in \text{SO}(d)$ is a random rotation matrix.

**Theorem 2.** *Assume $X_{\mathcal{T}}$ follows a $d$-dimensional multivariate Gaussian distribution $\mathcal{N}(\boldsymbol{\mu}, \Sigma)$. Let $X'_{\mathcal{T}}$ be the rotated representations of $X_{\mathcal{T}}$ such that:*

$$\lambda_{max}\left(\mathbb{E}\left[{X'_{\mathcal{T}}}^\top X'_{\mathcal{T}}\right]\right) \leq \lambda_{max}\left(\mathbb{E}\left[{X_{\mathcal{T}}}^\top X_{\mathcal{T}}\right]\right) \quad \Rightarrow \quad \sigma_{max}\left(\mathbb{E}\left[X'_{\mathcal{T}}\right]\right) \leq \sigma_{max}\left(\mathbb{E}\left[X_{\mathcal{T}}\right]\right), \quad (3)$$

*where $\lambda_{max}$ and $\sigma_{max}$ are the maximum eigenvalues and singular values, respectively.*

Theorem 2 states that random rotations reduce the maximum singular value of the data representations, implying that the principle component of $X_{\mathcal{T}}$ is weakened. In this case, the synthetic data cannot effectively capture the distribution of the real data, degenerating model performance. All proofs can be seen in Appendix A.

## 4 THE PROPOSED METHOD

### 4.1 PLUG-AND-PLAY POINT CLOUD ROTATOR

The above analysis demonstrates that learning rotation-invariant representations is crucial for point cloud distillation. However, a considerable part of point cloud models do not have this property. To solve this problem, we propose a plug-and-play point cloud rotator to transform the point clouds into their canonical view, enabling all methods to learn rotation-invariant representations.

**Rotation-equivariant.** We can leverage the orthogonality of the rotation matrix to eliminate its influence, *i.e.*, $RR^\top = I$, where PCA is a typical method:

$$\frac{1}{n}\sum \left(PR - \overline{P}R\right)^\top \left(PR - \overline{P}R\right) = R^\top U \Lambda U^\top R \quad \Rightarrow \quad (PR)(R^\top U) = PU, \tag{4}$$

where $\overline{P}$ is the center of $P$ and $U$ represents the eigenvectors of the covariance matrix. Notably, the projection $R^\top U$ is equivariant to the rotation of coordinates, and therefore $(PR)(R^\top U) = PU$ is rotation-invariant. However, the eigenvectors have the *sign ambiguity* issue, *i.e.*, $-u_i$ is also a valid eigenvector. As a result, the canonical view $PU$ is not unique and has 8 ambiguities in 3D space (Xiao et al., 2020; Yu et al., 2020), *i.e.*, $PUQ = P[\pm u_1, \pm u_2, \pm u_3]$, where $\left\{Q \in \mathbb{R}^{3\times3} | Q_{ii} = \{1, -1\}, Q_{ij} = 0, \forall i \neq j\right\}$ is a random reflection matrix.

**Sign-invariant.** Our rotator $r : \mathbb{R}^{n\times3} \to \mathbb{R}^{n\times3}$ is mainly designed to improve the performance of PCA by solving the sign ambiguity problem. Specifically, the rotator aims to learn a sign-equivariant reflection matrix $\overline{Q}$ for each point cloud such that $PUQ \cdot \overline{Q} = PU$ is sign-invariant. Specifically, the rotator first lifts the scalar coordinates to the vector representations:

$$H = [\sin(\pm PU), \sin(\pm 2PU) \cdots \sin(\pm tPU)]^\top = [\sin(PU), \sin(2PU) \cdots \sin(tPU)]^\top Q, \tag{5}$$

where $HQ \in \mathbb{R}^{n\times t\times3}$ is the sign-equivariant representations, $\sin(\cdot)$ is the sine function and $t$ is the period of Fourier features. An average pooling is then applied on $H$ to learn the representations of the whole point cloud. Finally, a learnable vector $w \in \mathbb{R}^t$ is used to decode the reflection matrix. The overall architecture of the rotation is formulated as follows:

$$r(P) = PUQ \cdot \overline{Q} = PUQ \cdot \text{Sign}(w \cdot \text{Pool}(HQ)), \tag{6}$$

where "Sign" means the signs of a matrix. The reflection matrix $\overline{Q}$ has the same signs as $Q$ because the sinusoidal features, pooling function, and linear decoder preserve the sign information of $HQ$, which can solve the sign ambiguity and learn sign-invariant representations.

**Other methods.** There are different approaches to learning rotation-invariant representations, such as vector neruon (Deng et al., 2021) and graph features (Kim et al., 2020). However, these methods break the original point coordinates, which are not easy to incorporate with other models. On the other hand, some methods try to solve the sign ambiguity by using pooling (Yu et al., 2020) and attention (Xiao et al., 2020; Li et al., 2021) mechanisms, which is inefficient as they need to calculate the representations for all ambiguous views.

### 4.2 POINT-WISE GENERATOR

In addition to the orientation, point clouds often have different resolutions, which also need to be considered in the distillation process. Traditional DD methods update the synthetic data in an *explicit* way, *i.e.*, directly optimizing the fixed-size tensors, which is unsuitable for point cloud synthesizing. On the other hand, the implicit neural representation (INR) methods (Sitzmann et al., 2020; Park et al., 2019) show great potential in generating data with arbitrary resolutions (Chen et al., 2021; Singh et al., 2023). Generally, INR predicts the signals of given coordinates, but the coordinates of the synthetic point clouds are unknown.

**Point Denoising.** Our solution is to use INR as a generator $g : \mathbb{R} \to \mathbb{R}^3$, whose input is random noise and output is the coordinates of a point. This means that we treat the noise as a special continuous coordinate, and the generator is used to obtain "3D signals" by denoising the noise. It is also worth noting that the generator adopts a point-wise paradigm rather than an instance-wise generation. Therefore, the amount of points is the same as the sampled noise, which allows us to generate point

Figure 2: Illustration of DD3D for part segmentation task. Each noise is first pre-partitioned into different parts according to its value, *e.g.*, the noise within (0, 0.45) is marked as fuselage. Then the generator maps the noise into coordinates to match the global (shape) and local (part) information.

clouds with infinite resolution. We choose SIREN (Sitzmann et al., 2020) as the generator, which can be formulated as:

$$g = [\Phi_1 \circ \Phi_2 \circ \cdots \circ \Phi_L] W_P, \quad \Phi_i = \sin(z_i w_i + b_i), \tag{7}$$

where $L$ is the number of layers, $\circ$ denotes the cascade of neural networks, $\Phi_i$ is a multi-layer perceptron (MLP) with sine activation function in the $i$-th layer, $z_1 \sim \mathcal{U}(0, 1)$ is the input noise, and $W_P \in \mathbb{R}^{d \times 3}$ is the decoder to generate 3D coordinates of points. Notably, we use uniform distribution instead of Gaussian distribution, as INR needs the input to be normalized within $[0, 1]$.

**Conditional Modulating.** The implicit generator can synthesize point clouds with arbitrary resolution but lacks label information, which is crucial for DD because it concentrates on classification task (Yu et al., 2024; Lei & Tao, 2024). Therefore, we use a modulator $c : \mathbb{R}^d \to \mathbb{R}^d$, which is implemented as another cascaded MLP $\Psi$, to encode the label information and generate conditions for the point cloud generation:

$$c = \Psi_1 \circ \Psi_2 \circ \cdots \circ \Psi_L, \quad \Psi_i = \text{ReLU}\left(m_i w_i' + b_i'\right), \tag{8}$$

where $\text{ReLU}(\cdot) = \max(0, \cdot)$, $m_i \in \mathbb{R}^d$ denotes the conditional representations and $m_1$ is a one-hot matrix, representing the label information. Assume that there are $K$ classes in total, and each class has $N$ synthetic point clouds, then $m_1 \in \mathbb{R}^{KN}$ and $w_1' \in \mathbb{R}^{KN \times d}$. The conditional representations are then used to modulate each layer of the generator. The overall architecture is formulated as:

$$g \odot c = \left[(\Psi_1 \odot \Phi_1) \circ (\Psi_2 \odot \Phi_2) \circ \cdots \circ (\Psi_L \odot \Phi_L)\right] W, \tag{9}$$

where $\odot$ is the element-wise multiplication to modulate the frequency and phase of the features. For clarity, in the following sections, we use $g(z, i)$ to denote $g \odot c$ with the $i$-th condition.

### 4.3 DISTILLATION TASKS

Traditional DD methods mainly focus on the fundamental image classification task. To better evaluate the performance of 3D distillation, we not only conduct experiments on the basic shape classification task but also explore the challenging part segmentation task. Shape classification aims to assign each point cloud a label, emphasizing global information, while part segmentation predicts the label of each point, which is more fine-grained.

**Shape Classification.** The distillation objective of the shape classification task is defined as:

$$\mathcal{L}_{shape} = \sum_{k=1}^{K} \mathcal{D}\left(\nabla \mathcal{L}_{cls}\left(f_\theta \circ r(B_k^{\mathcal{S}}), Y_k^{\mathcal{S}}\right), \nabla \mathcal{L}_{cls}\left(f_\theta \circ r(B_k^{\mathcal{T}}), Y_k^{\mathcal{T}}\right)\right), \tag{10}$$

where $K$ is the total classes of shapes, $B_k^{\mathcal{T}}$ and $Y_k^{\mathcal{T}}$ are a batch of real training data and labels, $B_k^{\mathcal{S}} = g(z, k)$ denotes a batch of synthetic point clouds belonging to the $k$-th class, and $Y_k^{\mathcal{S}}$ represents the predefined synthetic labels.

**Part Segmentation.** In the segmentation task, each shape is divided into several parts, *e.g.*, an airplane can be divided into fuselage, wings, engine, and rear. Such fine-grained labels need to be predefined before distillation. Therefore, DD3D first partitions the noise into different parts based on its value and then feeds the noise into the generator and rotator sequentially to obtain the synthetic

data. Intuitively, the synthetic data should encode both the global shapes and local geometry of the real data. Therefore, we propose global and local matching to match the gradients of the entire shape and individual parts, respectively. A toy example is shown in Figure 2.

The distillation objective of global matching in the part segmentation task is defined as:

$$\mathcal{L}_{part} = \sum_{k=1}^{K} \mathcal{D}\left(\nabla \mathcal{L}_{seg}\left(f_\theta \circ r(B_k^{\mathcal{S}}), V_k^{\mathcal{S}}\right), \nabla \mathcal{L}_{seg}\left(f_\theta \circ r(B_k^{\mathcal{T}}), V_k^{\mathcal{T}}\right)\right), \tag{11}$$

where $V^{\mathcal{T}}$ and $V^{\mathcal{S}}$ represents the real and synthetic part labels, respectively. To match the gradient of a specific part $p$, we apply an element-wise mask $M_p$ on the segmentation labels, *i.e.*, $V^{\mathcal{T}} = V^{\mathcal{T}} \odot M_p$ and $V^{\mathcal{S}} = V^{\mathcal{S}} \odot M_p$, to avoid interference from the gradients of other parts. In practice, we calculate local and global gradient matching alternately to preserve information of shapes and parts. See Algorithms 1, 2, 3, and Appendix B for the algorithm diagrams and detailed descriptions.

---

**Algorithm 1** DD3D for classification

**Input:** Training dataset $\mathcal{T}$
**Output:** Model $f$, Rotator $r$, Generator $g$
1: **for** $k = 1, \cdots, K$ **do**
2:      Initialize $f, r, g \sim \theta_f, \theta_r, \theta_g$
3:      **repeat**
4:          Sample a batch $B_k^{\mathcal{T}}, Y_k^{\mathcal{T}} \sim \mathcal{T}$
5:          Sample noise $z_1 \sim \mathcal{U}(0, 1)$
6:          Generate $B_k^{\mathcal{S}} = g(z_1, k)W$
7:          Compute $\nabla \mathcal{L}_{cls}^{\mathcal{S}}$ and $\nabla \mathcal{L}_{cls}^{\mathcal{T}}$
8:          Update $\theta_g$ with $\mathcal{L}_{shape}$
9:          **repeat**
10:             Update $\theta_f, \theta_r$ with $\mathcal{L}_{cls}^{\mathcal{S}}$
11:          **until** inner-loop end
12:      **until** outer-loop end
13: **end for**

---

**Algorithm 2** DD3D for evaluation

1: Generate $B^{\mathcal{S}}, Y^{\mathcal{S}}$ or $V^{\mathcal{S}}$ via $g$
2: Optimize $\theta_f$ and $\theta_r$ until convergence
3: Evaluate $f \circ r$ on the test dataset

---

**Algorithm 3** DD3D for part segmentation

**Input:** Training dataset $\mathcal{T}$
**Output:** Model $f$, Rotator $r$, Generator $g$
1: **for** $k = 1, \cdots, K$ **do**         ▷ Shape Classes
2:      Initialize $f, r, g \sim \theta_f, \theta_r, \theta_g$
3:      **repeat**
4:          Sample a batch $B_k^{\mathcal{T}}, V_k^{\mathcal{T}} \sim \mathcal{T}$
5:          Sample noise $z_1 \sim \mathcal{U}(0, 1)$
6:          Generate $V_k^{\mathcal{S}}$ by partitioning noise
7:          Generate $B_k^{\mathcal{S}} = g(z_1, k)W$
8:          Compute $\nabla \mathcal{L}_{seg}^{\mathcal{S}}$ and $\nabla \mathcal{L}_{seg}^{\mathcal{T}}$ ▷ Shape Info.
9:          **for** $p \in k$ **do**      ▷ Part belongs to a shape
10:             Apply mask $M_p$ on $V_k^{\mathcal{S}}$
11:             Compute $\nabla \mathcal{L}_{seg}^{\mathcal{S}}$ and $\nabla \mathcal{L}_{seg}^{\mathcal{T}}$ ▷ Part Info.
12:          **end for**
13:          Update $\theta_g$ with $\mathcal{L}_{part}$
14:          **repeat**
15:             Update $\theta_f, \theta_r$ with $\mathcal{L}_{seg}^{\mathcal{S}}$
16:          **until** inner-loop end
17:      **until** outer-loop end
18: **end for**

---

## 4.4 DISCUSSION

**Complexity.** The complexity of DD contains three parts: data parameterization, model forward, and data alignment. For a point cloud with $n$ points, DD3D has an additional time complexity $\mathcal{O}(Lnd)$ to generate synthetic data, which makes the time overhead higher than that of vanilla DD methods. But if we consider down-sampling the points, the time complexity of all three parts can be reduced. See Section 5.5 for a comprehensive comparison of the time and space overhead between DD and DD3D.

**Limitations.** Unlike parameterizing data as an explicit matrix, DD3D leverages a generator to synthesize data, significantly reducing the computational costs and memory budget. However, a major drawback is that the generator cannot take original data as initialization, which may affect the convergence of the model. A comparison can be found in Section 5.4. Moreover, there are still some issues that remain unsolved. For example, existing methods cannot applied to tasks with continuous labels, such as detection, which limits their applications. Besides, how to make the synthetic datasets learn fine-grained details beyond shapes remains a challenge.

## 5 EXPERIMENTS

We benchmark our method on two fundamental tasks of point cloud analysis: shape classification (Section 5.1) and part segmentation (Section 5.2), followed by a series of analyses, including generalization (Section 5.3), ablation (Section 5.4), and visualization (Section 5.6).

**Datasets.** We employ three datasets of different scales for the shape classification task: ($i$) ScanObjectNN (*OBJ_BG*) (Uy et al., 2019) is the smallest dataset but consists of real-world data, which is challenging to distillate. ($ii$) ModelNet40 (Wu et al., 2015) is a larger synthetic dataset generated from CAD models. ($iii$) MVPNet (Yu et al., 2023) is the largest dataset, containing 87K point clouds scanned from real-world videos. We use its subset MVPNet100, which includes data from the 100 most populous categories, to alleviate the influence of long-tail distribution, similar to the CAFIR-100 dataset[2]. For the part segmentation task, we follow Qi et al. (2017a) and choose ShapeNet-part (Yi et al., 2016) dataset for evaluation. All the datasets use the standard data splits, and their detailed statistic information can be found in Appendix C.

**Data Preparation and Metrics.** Each cloud contains 1,024 points and is normalized into a unit sphere. We consider two settings: *Aligned* and *Rotated*. In the *Aligned* setting, both training and test point clouds have the same orientation, while in the *Rotated* setting, both training and test data are rotated randomly. For the rotated data, we project them along the direction of maximum variance during pre-processing. Note that the point clouds in MVPNet only have 180° views, so we do not randomly rotate them. The details of pre-processing can be found in Appendix C. We report the Overall Accuracy (OA, %) of each method in the shape classification task and the average class intersection of union (IoU, %) in the part segmentation task.

**Baselines.** To demonstrate the effectiveness of our method, we choose two types of baselines: (1) Coreset-based methods, including Random, Herding (Welling, 2009) and K-Center (Sener & Savarese, 2018). (2) Distillation-based methods, including Gradient Matching (GM) (Zhao et al., 2021), Distribution Matching (DM) (Zhao & Bilen, 2023), and Trajectory Matching (TM) (Cazenavette et al., 2022). We choose GM as the distillation objection for DD3D as it makes a trade-off between time and memory consumption. See Appendix D for the detailed hyperparameters.

**Backbones.** We provide a lightweight PointNet as the distillation backbone, which abandons the transformation network because previous literature (Yu et al., 2024) pointed out that complex network architecture may lead to degraded distillation performance. See Appendix E for more details. In the evaluation stage, we adopt various advanced backbones to evaluate the generalization ability of distilled datasets, including PointNet++ (Qi et al., 2017b), DGCNN (Wang et al., 2019), Point Transformer (Guo et al., 2021a), PointMLP (Ma et al., 2022), and PointNext (Qian et al., 2022). Results can be found in Table 3.

**Experimental Setup.** For each method, we perform the distillation process twice, evaluate each synthetic point cloud dataset five times (10 results in total), and report the mean and standard deviation. Baselines are all initialized with original data, while DD3D is trained from scratch. For the shape classification task, we consider three different distillation ratios with 1, 10, and 50 synthetic point clouds per class (CPC). For the part segmentation task, we only choose CPC=1 due to the limitation of GPU memory.

## 5.1 SHAPE CLASSIFICATION

The results of different methods on the shape classification task are shown in Table 1, from which we have the following observations. Firstly, the results of distillation-based methods consistently outperform coreset-based methods, demonstrating the effectiveness of DD. However, as the amount of synthetic data increases, the performance of the coreset increases rapidly. Secondly, DD3D achieves state-of-the-art performance on all five datasets, demonstrating its superiority over traditional DD methods. Notably, DD3D obtains more improvements over baselines as the number of CPCs increases, possibly because the generator provides more diverse data. Thirdly, the results on the rotated data are weaker than those on the aligned data. Although we project the rotated data to the canonical orientation, *i.e.*, direction with maximum variance, these point clouds still have slightly different orientations, while the aligned data is manually registered, which is strictly towards the direction of gravity and therefore has better performance.

## 5.2 PART SEGMENTATION

We illustrate the results of the part segmentation task in Table 2. As the segmentation task is different from the basic classification task, some coreset and distillation methods cannot adapt to it. Therefore,

---

[2]https://www.cs.toronto.edu/~kriz/cifar.html

Table 1: Shape classification results of different methods, mean accuracy (%) $\pm$ standard deviation. **Bold** indicates the best performance, and "-" means out-of-memory during distillation. CPC: Number of Clouds Per Class.

| Dataset | CPC | Coreset-based | | | Distillation-based | | | | Full Dataset |
|---|---|---|---|---|---|---|---|---|---|
| | | Random | Herding | K-Center | GM | DM | TM | DD3D | |
| ScanObjectNN (Aligned) | 1 | 22.00±2.56 | 16.29±1.37 | 18.18±1.04 | 26.34±2.07 | 25.90±1.34 | 26.42±2.08 | **30.62±1.75** | 66.96 |
| | 10 | 32.63±1.51 | 31.94±3.31 | 33.46±1.46 | 39.87±3.00 | 37.61±2.78 | 36.44±2.74 | **43.77±2.63** | |
| | 50 | 54.15±1.77 | 51.70±1.87 | 54.22±1.30 | 57.52±2.03 | 56.91±1.17 | - | **61.96±1.44** | |
| ScanObjectNN (Rotated) | 1 | 14.90±2.10 | 18.10±1.55 | 19.91±2.16 | 14.64±3.04 | 18.74±2.44 | 19.29±3.90 | **23.59±2.17** | 54.84 |
| | 10 | 20.50±1.26 | 20.20±2.19 | 22.05±1.76 | 20.55±3.99 | 20.26±4.31 | 19.20±4.52 | **25.84±3.11** | |
| | 50 | 42.98±1.84 | 43.39±1.34 | 44.29±2.07 | 47.74±1.82 | 48.11±2.30 | - | **50.26±1.42** | |
| ModelNet40 (Aligned) | 1 | 40.53±0.36 | 43.41±0.81 | 43.90±1.51 | 53.38±0.86 | 53.21±0.58 | 52.37±0.99 | **53.82±0.28** | 88.05 |
| | 10 | 71.89±0.29 | 74.63±0.48 | 73.13±0.78 | 75.45±0.82 | 74.45±0.47 | 75.39±1.32 | **76.31±0.49** | |
| | 50 | 82.37±0.45 | 82.75±0.49 | 82.73±0.28 | 81.74±0.55 | 83.02±1.16 | - | **83.91±0.23** | |
| ModelNet40 (Rotated) | 1 | 34.65±0.71 | 30.03±1.42 | 30.05±0.50 | 41.32±1.96 | 41.71±1.65 | 37.36±2.98 | **42.36±0.83** | 80.45 |
| | 10 | 58.87±0.65 | 56.03±0.62 | 57.69±0.97 | 55.69±1.63 | 55.45±1.80 | 56.21±1.14 | **58.14±1.36** | |
| | 50 | 70.13±0.64 | 70.02±0.71 | 69.68±0.59 | 68.92±0.73 | 69.31±0.79 | - | **71.27±0.32** | |
| MVPNet100 | 1 | 5.21±0.27 | 8.14±0.22 | 8.41±0.35 | 10.52±0.83 | 11.73±0.49 | 10.74±0.57 | **13.68±0.48** | 55.63 |
| | 10 | 15.99±0.30 | 22.11±0.21 | 20.54±0.21 | 25.68±0.77 | 25.71±0.69 | - | **31.14±1.31** | |
| | 50 | 30.14±0.27 | 35.87±0.24 | 35.48±0.44 | 37.41±0.57 | 36.83±0.20 | - | **40.61±0.38** | |

*Note*: All methods with rotated data are trained with the point cloud rotator. Ablations can be seen in Table 5.

Table 2: Part Segmentation results on the ShapeNet dataset, mean IoU (%).

| | mIoU | airplane | bag | cap | car | chair | earphone | guitar | knife | lamp | laptop | motor | mug | pistol | rocket | skateboard | table |
|---|---|---|---|---|---|---|---|---|---|---|---|---|---|---|---|---|---|
| Full | 74.43 | 77.06 | 74.88 | 69.26 | 75.27 | 76.16 | 69.89 | 78.22 | 76.66 | 74.72 | 77.03 | 73.49 | 73.84 | 78.03 | 74.03 | 66.54 | 75.73 |
| Coreset | 48.83 | 47.03 | 24.68 | 58.89 | 39.57 | 70.13 | 30.78 | 74.15 | 58.46 | 42.24 | 89.34 | 26.78 | 37.93 | 56.01 | 20.18 | 41.59 | 63.41 |
| GM | 47.94 | 48.93 | 20.34 | 42.12 | 29.98 | 73.06 | 24.22 | 73.41 | 69.24 | 32.38 | 83.80 | 20.96 | 61.58 | 44.17 | 38.62 | 43.32 | 60.84 |
| DD3D | **50.99** | 42.39 | 34.37 | 54.00 | 29.20 | 70.52 | 27.87 | 77.16 | 74.83 | 34.09 | 86.52 | 28.46 | 64.93 | 53.04 | 34.89 | 43.62 | 59.94 |

we only compare DD3D with the random coreset selection and gradient matching methods. It can be observed that the performance of GM is not as good as the random coreset method, although it is initialized by the real data. On the other hand, DD3D outperforms both methods, validating its advantages in learning the coordinates and labels of point clouds. However, the performance of DD3D is not as good as the full dataset because part segmentation needs to learn both global information, *i.e.*, shapes, and local information, *i.e.*, parts, which is a challenging task in 3D distillation. The visualizations of DD3D with different matching objectives can be seen in Appendix F.

## 5.3 GENERALIZATION STUDIES

We conduct two generalization experiments to verify the effectiveness of DD3D further.

**Cross-architecture Generalization.** We first evaluate whether DD3D can benefit different point cloud models. Specifically, we use PointNet as the distillation method and utilize five advanced point cloud models as evaluation methods, trained on the synthetic data from scratch. Notably, we use synthetic data with CPC=50 to alleviate the randomness. The results are shown in Table 3, from which we can see that DD3D consistently outperforms DM and GM across different datasets and evaluation methods, proving that the synthetic data distilled by DD3D has better generalizability. This may be attributed to the generator that provides various point clouds in each epoch by sampling different noises, which plays a role like data augmentation. However, we can also observe that the results of evaluation methods are not as good as PointNet, emphasizing that the synthetic data is still biased by the distillation model.

**Cross-resolution Generalization.** We next explore the performance of DD3D under different resolutions. Typically, the shape classification task needs 1,024 points for training and evaluation. In this experiment, we randomly sample 256 and 512 points from real data to supervise the distillation of DD3D. Once trained, we leverage DD3D to generate 1,024 points for evaluation. It is visible from Figure 3 that training on high-resolution point clouds can accelerate the convergence of DD3D but the final matching losses are similar. Moreover, Table 4 shows that different resolutions have similar performance. In some cases, low-resolution data also outperforms high-resolution point clouds, *e.g.*, ScanObjectNN. This discovery shows that DD3D can not only achieve stable results but also significantly reduce computational costs and GPU occupancy.

Table 3: Cross-architecture results (%) on different datasets with CPC=50.

| Dataset | Ratio | Method | PointNet++ | DGCNN | PCT | PointMLP | PointNeXt | Average |
|---|---|---|---|---|---|---|---|---|
| ScanObjectNN | 32.3% | DM | 56.02 | 51.47 | 52.72 | 51.33 | 51.82 | 52.67 |
| | | GM | 55.38 | 52.98 | 53.28 | 51.33 | 52.81 | 53.16 |
| | | DD3D | **57.14** | **53.36** | **54.04** | **52.50** | **53.36** | **54.08** |
| ModelNet40 | 20.3% | DM | 74.35 | 74.84 | 76.92 | 72.49 | 71.48 | 74.02 |
| | | GM | 76.54 | 73.38 | 77.31 | 74.11 | 72.00 | 74.67 |
| | | DD3D | **77.71** | **75.36** | **79.21** | **75.36** | **73.99** | **76.33** |
| MVPNet100 | 8.0% | DM | 33.20 | 31.26 | 33.92 | 32.58 | 31.17 | 32.43 |
| | | GM | 31.35 | 29.88 | 31.43 | 31.79 | 30.82 | 31.09 |
| | | DD3D | **34.19** | **32.94** | **35.82** | **33.08** | **32.75** | **33.76** |

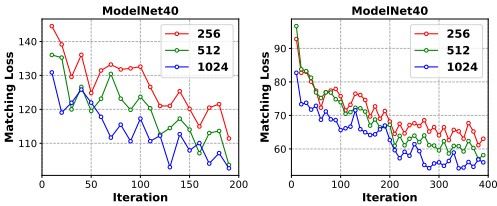

Figure 3: Matching loss of different resolutions.

Table 4: DD3D under different resolutions.

| CPC=50 | Resolution | | | |
|---|---|---|---|---|
| | 256 | 512 | 1024 | Avg. |
| ScanObjectNN | 61.27 | 60.59 | **61.96** | 61.27 |
| ModelNet40 | 83.03 | 83.59 | **83.91** | 83.51 |
| MVPNet100 | 39.88 | 40.13 | **40.61** | 40.21 |

## 5.4 ABLATION STUDIES

**Point Cloud Rotator.** We first verify the effectiveness of the proposed point cloud rotator on the rotated ModelNet40 dataset. Specifically, we consider three different models: (1) PointNet, which is rotation-sensitive; (2) PointNet + PCA, which is rotation-invariant but sign-variant; (3) PointNet + Rotator, which is rotation- and sign-invariant. It can be observed from Table 5 that the performance of all methods drops rapidly when the data is randomly rotated. On the other hand, leveraging PCA to transform the point clouds into a canonical orientation can significantly improve the distillation performance. However, the results are still far from the model with the point cloud rotator, which reflects that sign ambiguity will seriously prevent the distillation model from learning meaningful synthetic data. Finally, it can be observed that the proposed rotator can help point cloud models to rotation-invariant representations, thus benefiting the learning of synthetic data.

**Point-wise Generator.** In Section 4.4, we have discussed the pros and cons of DD and DD3D. Here, we make a further attempt to show the advantages of the proposed point-wise generator. Firstly, in Table 6, we report the results of DD with and without initialization. It is noticeable that initializing DD with real data is important for the distillation performance. However, its performance is still not as good as DD3D, which does not rely on any initialization. Moreover, the performance of DD3D can still be improved if we use sampling during the evaluation, *i.e.*, generating different point clouds at each epoch, because the generated data serves as data augmentation to improve the model performance. This strategy is more useful when the value of CPC is small. Additionally, in Figure 4, we visualize the matching loss of DD and DD3D. It is observable that DD without initialization has a higher loss value and converges more slowly than DD3D, reflecting the advantages of the proposed point-wise generator in synthetic point clouds.

## 5.5 TIME AND SPACE OVERHEAD

We compare the overhead between DD and DD3D from multiple views. Firstly, Figure 8a shows that the time overhead of DD3D is slightly higher than DD due to the generation of synthetic data. Then, we can observe from Figure 8b that the memory budget of DD grows faster than DD3D as the value of CPC increases. DD3D can save the budget of synthetic data by sharing the generator between different classes, and its memory is nearly 4x smaller than DD when CPC=10. Figure 4c illustrates the changes in time and space overhead of DD3D at different resolutions. We can see that training with low-resolution point clouds significantly reduces overhead, which is important for resource-constrained scenarios, such as edge computing.

Table 5: Ablation studies of the point cloud rotator on the ModelNet40 dataset with CPC=50.

| ModelNet40 | Random | GM | DM | DD3D |
|---|---|---|---|---|
| PointNet | 14.75 | 9.47 | 10.16 | 17.91 |
| PointNet + PCA | 60.77 | 53.55 | 55.57 | 62.72 |
| PointNet + Rotator | 70.13 | 68.92 | 69.31 | 71.27 |
| Full Dataset | | 80.45 | | |

Table 6: Comparison between DD and DD3D with different training strategies.

| ModelNet40 | 1 | 10 | 50 |
|---|---|---|---|
| DD w/o Initialization | 53.08 | 72.57 | 81.77 |
| DD w/ Initialization | 53.38 | 75.45 | 83.02 |
| DD3D w/o Sampling | 53.82 | 76.01 | 83.72 |
| DD3D w/ Sampling | **54.27** | **76.15** | **83.91** |

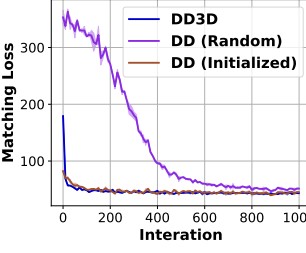

Figure 4: Matching loss of methods with different settings.

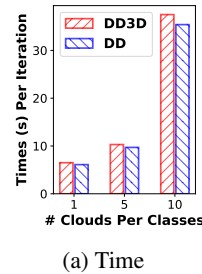
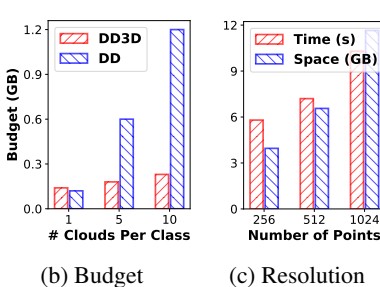

(a) Time    (b) Budget    (c) Resolution

Figure 5: Time and space overhead between DD and DD3D.

## 5.6 VISUALIZATION

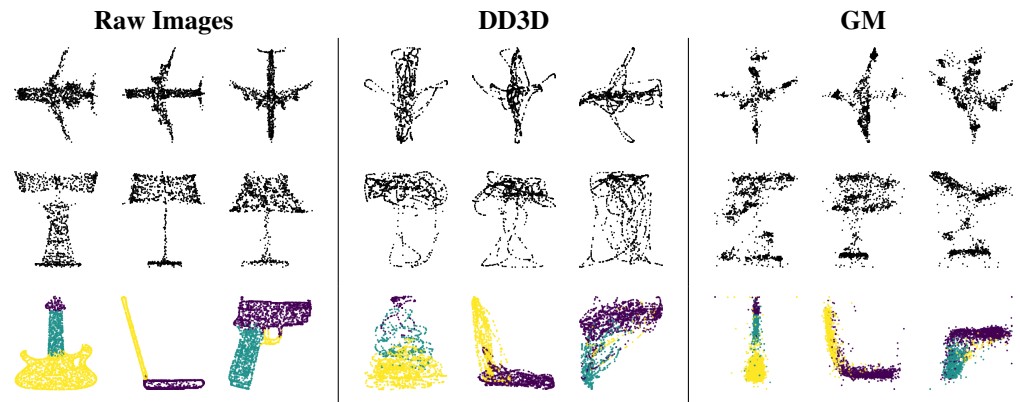

Figure 6: Visualization of the real and synthetic datasets. Top row: ModelNet40 (Airplane). Middle row: ModelNet40 (Lamp). Bottom row: ShapeNet (Guitar, Laptop, and Pistol).

We visualize the real and synthetic point clouds in Figure 6 for a more intuitive comparison. The results of DD3D and GM are placed in the last two columns. It can be observed that the point clouds generated by GM tend to condense to some clusters, while some isolated points are left as noise. On the contrary, the point clouds generated by DD3D are coherent and encode the global geometric shapes. Moreover, in ShapeNet, the point clouds of GM are squeezed, making its shape inconsistent with the real dataset, while the results of DD3D are more realistic and encode the spatial relationship between parts, validating the effectiveness of DD3D for 3d data. See Appendix F for more visualizations.

## 6 CONCLUSION

This paper introduces DD3D for 3D point cloud distillation, which matches the rotation-invariant data distribution between real and synthetic data by transforming point clouds into a canonical orientation. Once trained, DD3D can synthesize point clouds at arbitrary resolutions, reducing memory budget and improving scalability. Extensive experiments on both classification and segmentation tasks validate the superiority of DD3D over traditional DD methods. A promising direction is to initialize DD3D with real data to improve its performance.

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

## A  PROOF OF THEOREMS

**Theorem 1.** *Assume the classifier is a linear layer $W$ and $\mathcal{L}_{cls}$ can be simplified to the mean-squared error $\|XW - Y\|_F^2$. The objective of gradient matching is equal to variance preserving:*

$$\min_{\mathcal{S}} \mathcal{L}_{GM} = \min_{\mathcal{S}} \mathcal{D}\left(\nabla_W \mathcal{L}_{cls}^{\mathcal{S}}, \nabla_W \mathcal{L}_{cls}^{\mathcal{T}}\right) \quad \Rightarrow \quad \min_{\mathcal{S}} \left\|X_{\mathcal{S}}^{\top} X_{\mathcal{S}} - X_{\mathcal{T}}^{\top} X_{\mathcal{T}}\right\|_F^2, \quad (12)$$

*where $\mathcal{D}$ is a distance metric and $\nabla_W$ is the gradient with respect to $W$.*

*Proof.* The gradient of $\|XW - Y\|_F^2$ is denoted as $\nabla = X^{\top}(XW - Y)$. We can then match the gradients between the real and synthetic data:

$$\|\nabla_{\mathcal{S}} - \nabla_{\mathcal{T}}\|_F^2 = \|X_{\mathcal{S}}^{\top}(X_{\mathcal{S}}W - Y_{\mathcal{S}}) - X_{\mathcal{T}}^{\top}(X_{\mathcal{T}}W - Y_{\mathcal{T}})\|_F^2 \quad (13)$$

$$\leq \|W\|_F^2 \underbrace{\|X_{\mathcal{S}}^{\top} X_{\mathcal{S}} - X_{\mathcal{T}}^{\top} X_{\mathcal{T}}\|_F^2}_{\text{Variance}} + \underbrace{\|X_{\mathcal{S}}^{\top} Y_{\mathcal{S}} - X_{\mathcal{T}}^{\top} Y_{\mathcal{T}}\|_F^2}_{\text{Mean}}. \quad (14)$$

We can see that the first term is to preserve the variance of real data, and the second term aligns the average representations of samples belonging to the same class. These two terms can be combined if we set $\tilde{X}_{\mathcal{S}} = X_{\mathcal{S}} - X_{\mathcal{S}}^{\top} Y_{\mathcal{S}}$ and $\tilde{X}_{\mathcal{T}} = X_{\mathcal{T}} - X_{\mathcal{T}}^{\top} Y_{\mathcal{T}}$ for each class. Then we only need to match the variance between $\tilde{X}_{\mathcal{S}}$ and $\tilde{X}_{\mathcal{T}}$. □

**Theorem 2.** *Assume $X_{\mathcal{T}}$ follows a $d$-dimensional multivariate Gaussian distribution $\mathcal{N}(\boldsymbol{\mu}, \Sigma)$. Let $X_{\mathcal{T}}'$ be the rotated representations of $X_{\mathcal{T}}$ such that:*

$$\lambda_{max}\left(\mathbb{E}\left[{X_{\mathcal{T}}'}^{\top} X_{\mathcal{T}}'\right]\right) \leq \lambda_{max}\left(\mathbb{E}\left[{X_{\mathcal{T}}}^{\top} X_{\mathcal{T}}\right]\right) \quad \Rightarrow \quad \sigma_{max}\left(\mathbb{E}\left[X_{\mathcal{T}}'\right]\right) \leq \sigma_{max}\left(\mathbb{E}\left[X_{\mathcal{T}}\right]\right), \quad (15)$$

*where $\lambda_{max}$ and $\sigma_{max}$ are the maximum eigenvalues and singular values, respectively.*

*Proof.* Firstly, the largest eigenvalue of the covariance matrix ${X_{\mathcal{T}}}^{\top} X_{\mathcal{T}}$ is equal to the largest singular value of $X_{\mathcal{T}}$. Therefore, we only prove the first inequality.

Secondly, as $X^{\top} X = \sum_{i=1}^{n} x_i^{\top} x_i$, for $X_{\mathcal{T}} \sim \mathcal{N}(\boldsymbol{\mu}, \Sigma)$, we have:

$$\mathbb{E}\left[X_{\mathcal{T}}^{\top} X_{\mathcal{T}}\right] = \mathbb{E}\left[\sum_{i=1}^{n} x_i^{\top} x_i\right] = \sum_{i=1}^{n} \mathbb{E}\left[x_i^{\top} x_i\right] = n\left(\boldsymbol{\mu}^{\top} \boldsymbol{\mu} + \Sigma\right), \quad (16)$$

$$\mathbb{E}\left[{X_{\mathcal{T}}'}^{\top} X_{\mathcal{T}}'\right] = \mathbb{E}\left[\sum_{i=1}^{n} R_i^{\top} x_i^{\top} x_i R_i\right] = \sum_{i=1}^{n} \mathbb{E}\left[R_i^{\top} x_i^{\top} x_i R_i\right] = \sum_{i=1}^{n} R_i^{\top} \mathbb{E}\left[x_i^{\top} x_i\right] R_i. \quad (17)$$

Thirdly, we have:

$$\lambda_{max}\left(\mathbb{E}\left[X_{\mathcal{T}}^{\top} X_{\mathcal{T}}\right]\right) = n\lambda_{max}\left(\boldsymbol{\mu}^{\top} \boldsymbol{\mu} + \Sigma\right), \quad (18)$$

$$\lambda_{max}\left(\mathbb{E}\left[{X_{\mathcal{T}}'}^{\top} X_{\mathcal{T}}'\right]\right) = \lambda_{max}\left(\sum_{i=1}^{n} R_i^{\top} \mathbb{E}\left[x_i^{\top} x_i\right] R_i\right) \leq \sum_{i=1}^{n} \lambda_{max}\left(R_i^{\top} \mathbb{E}\left[x_i^{\top} x_i\right] R_i\right) \quad (19)$$

$$= \sum_{i=1}^{n} \lambda_{max}\left(R_i^{\top} \boldsymbol{\mu}^{\top} \boldsymbol{\mu} R_i + R_i^{\top} \Sigma R_i\right) \leq \lambda_{max}\left(\mathbb{E}\left[X_{\mathcal{T}}^{\top} X_{\mathcal{T}}\right]\right). \quad (20)$$

The above inequality shows that the largest eigenvalue of $\mathbb{E}\left[X_{\mathcal{T}}^{\top} X_{\mathcal{T}}\right]$ is the upper bound of $\mathbb{E}\left[{X_{\mathcal{T}}'}^{\top} X_{\mathcal{T}}'\right]$. The equality holds if and only if the random rotation matrices are commutative, which is infeasible in practice. □

## B  IMPLEMENTATION DETAILS OF DD3D

Here, we explain some details of DD3D, consisting of two important components: a point cloud rotator and a point-wise generator. Both components are built based on the SIREN (Sitzmann et al., 2020) model, which stacks multiple fully connected layers with $\sin(\cdot)$ activation to capture the high-frequency information. The PyTorch code is shown in Algorithm 4, where some details are highlighted.

**Algorithm 4** PyTorch code of DD3D

```python
import torch
import torch.nn as nn
import SIREN

class Rotator(nn.Module):
    def __init__(self, hidden_dim, w0):
        super().__init__()

        # w0 is to adjust the frequency of sine function
        self.sign_encoder = SIREN(1, hidden_dim, w0=w0)
        self.sign_decoder = SIREN(hidden_dim, 1, w0=1.)

    def forward(self, x):
        x = x.unsqueeze(-1) # x: [B, N, 3, 1]

        feat = self.sign_encoder(x).mean(dim=1, keepdim=True) # [B, N, 3, 1] -> [B, 1, 3, d]
        feat = self.sign_decoder(feat) # [B, 1, 3, d] -> [B, 1, 3, 1]
        sign = torch.sign(feat) # sign-equivariant

        x = x * sign # [B, N, 3, 1] * [B, 1, 3, 1] -> [B, N, 3, 1]
        return x.squeeze(-1)
```

## C    DETAILS OF DATASETS

Table 7: Details of datasets

|  | ScanObjectNN | ModelNet40 | MVPNet | ShapeNet |
|---|---|---|---|---|
| # Shape Classes | 15 | 40 | 100 | 16 |
| # Part Classes | - | - | - | 50 |
| # Training Samples | 2,322 | 9,843 | 62,494 | 14,007 |
| # Validation Samples | 580 | 2,468 | 15,670 | 2,874 |
| Resolution | 1,024 | 1,024 | 1,024 | 2,048 |

The detailed statistical information of the datasets used in this paper is shown in Table 7. We list the sources of the datasets and their licenses in the following.

- **ScanObjectNN**: https://github.com/feiran-l/rotation-invariant-pointcloud-analysis (MIT license)

- **ModelNet40**: http://modelnet.cs.princeton.edu/ModelNet40.zip

- **MVPNet**: https://github.com/GAP-LAB-CUHK-SZ/MVImgNet

- **ShapeNet**: https://github.com/feiran-l/rotation-invariant-pointcloud-analysis (MIT license)

## D    HYPERPARAMETERS

The hyperparameters of baselines and DD3D are listed in Tables 8 and 9, respectively.

Table 8: Hyperparameters used for Data Synthesis.

|  | ScanObjectNN | ModelNet40 | MVPNet100 | ShapeNet |
|---|---|---|---|---|
| Optimizer | Adam | Adam | Adam | Adam |
| Initial LR | 0.001 | 0.001 | 0.001 | 0.001 |
| Batch Size | 32 | 32 | 64 | 32 |
| Iterations | 200 | 400 | 600 | 200 |
| Weight Decay | 0.0005 | 0.0005 | 0.0005 | 0.0005 |
| Augmentation | Scale, Jitter, Rotate | Scale, Jitter, Rotate | Scale, Jitter, Rotate | Scale, Jitter, Rotate |
| Scheduler | StepLR (Decay 0.1 / 100 iter) | StepLR (Decay 0.1 / 100 iter) | StepLR (Decay 0.5 / 200 iter) | - |

Table 9: Hyperparameters used for Validation.

| | ScanObjectNN | ModelNet40 | MVPNet100 | ShapeNet |
|---|---|---|---|---|
| Optimizer | Adam | Adam | Adam | Adam |
| Initial LR | 0.001 | 0.001 | 0.001 | 0.001 |
| Batch Size | 8 | 8 | 32 | 8 |
| Epochs | 200 | 200 | 200 | 200 |
| Weight Decay | 0.0005 | 0.0005 | 0.0005 | 0.0005 |
| Augmentation | Scale, Jitter, Rotate | Scale, Jitter, Rotate | Scale, Jitter, Rotate | Scale, Jitter, Rotate |
| Scheduler | StepLR (Decay 0.1 / 100 epoch) | StepLR (Decay 0.1 / 100 epoch) | CosineAnnealingLR | - |

## E DETAILS OF BACKBONES

Previous work (Yu et al., 2024) pointed out that complexity architecture may degenerate the distillation performance. Therefore, we re-implement some traditional point cloud models, described as follows.

- **PointNet**: 3 layers with dimension [64, 128, 1024]. Each layer consists of a Conv1d layer, an InstanceNorm1d or BatchNorm1d layer (for classification and segmentation, respectively), and a ReLU activation. We use max-pooling to learn the global representation of point clouds.

For other methods, we use the codes provided by openpoints[3] library. The YAML configuration files are listed below.

| PoineNet++ | DGCNN | PCT | PointMLP | PointNext |
|---|---|---|---|---|

```
model:
  NAME: BaseCls
  encoder_args:
    NAME: PointNet2Encoder
    in_channels: 3
    width: null
    layers: 3
    use_res: False
    strides: [2, 4, 1]
    mlps: [[[64, 128]],
           [[128, 256]],
           [[256, 1024]]]
    radius: [0.2, 0.4, null]
    num_samples: [32, 64, null]
    sampler: fps
    aggr_args:
      NAME: 'convpool'
      feature_type: 'dp_fj'
      anisotropic: False
      reduction: 'max'
    group_args:
      NAME: 'ballquery'
      use_xyz: True
      normalize_dp: False
    conv_args:
      order: conv-norm-act
    act_args:
      act: 'relu'
    norm_args:
      norm: 'bn'
  cls_args:
    NAME: ClsHead
    num_classes: 40
    mlps: [256]
    norm_args:
      norm: 'bn1d'
```

```
model:
  NAME: BaseCls
  encoder_args:
    NAME: DGCNN
    in_channels: 3
    channels: 64
    n_classes: 40
    emb_dims: 512
    n_blocks: 5
    conv: 'edge'
    k: 20
    dropout: 0.5
    norm_args: {'norm': 'bn'}
    act_args: {'act': 'leakyrelu'}
  cls_args:
    NAME: ClsHead
    num_classes: 15
    mlps: [256]
    norm_args:
      norm: 'bn1d'
```

```
num_point: 1024
num_class: 40
input_dim: 3
model:
  nneighbor: 16
  nblocks: 1
  transformer_dim: 256
```

```
model:
  NAME: PointMLP
  in_channels: 3
  points: 1024
  num_classes: 40
  embed_dim: 64
  groups: 1
  res_expansion: 1.0
  activation: "relu"
  bias: False
  use_xyz: False
  normalize: "anchor"
  dim_expansion: [ 2]
  pre_blocks: [ 2]
  pos_blocks: [ 2]
  k_neighbors: [ 24]
  reducers: [ 2]
```

```
model:
  NAME: BaseCls
  encoder_args:
    NAME: PointNextEncoder
    blocks: [1, 1, 1, 1, 1, 1]
    strides: [1, 2, 2, 2, 2, 1]
    width: 32
    in_channels: 3
    radius: 0.15
    radius_scaling: 1.5
    sa_layers: 2
    sa_use_res: True
    nsample: 32
    expansion: 4
    aggr_args:
      feature_type: 'dp_fj'
      reduction: 'max'
    group_args:
      NAME: 'ballquery'
      normalize_dp: True
    conv_args:
      order: conv-norm-act
    act_args:
      act: 'relu'
    norm_args:
      norm: 'bn'
  cls_args:
    NAME: ClsHead
    num_classes: 40
    mlps: [256]
    norm_args:
      norm: 'bn1d'
```

## F ADDITIONAL EXPERIMENTS AND DISCUSSIONS

**Performance of Synthetic Datasets.** The performance of DD is positively correlated with the memory overhead, *i.e.*, CPC, of the synthetic datasets. When applied to applications requiring high accuracy, we can increase the value of CPC to improve the performance of synthetic datasets. To validate this, we conduct experiments on ScanObjectNN and ModelNet40 with CPC=100 and evaluate the performance of DD3D and GM. The results are shown below.

Table 10: Results on ScanObjectNN

| CPC | 1 | 10 | 50 | 100 |
|---|---|---|---|---|
| GM | 26.34 | 39.87 | 57.52 | 62.82 |
| DD3D | 30.62 | 43.77 | 61.96 | 65.51 |
| Full | | 66.96 | | |

Table 11: Results on ModelNet40

| CPC | 1 | 10 | 50 | 100 |
|---|---|---|---|---|
| GM | 53.38 | 65.45 | 81.74 | 84.71 |
| DD3D | 53.82 | 76.31 | 83.91 | 86.68 |
| Full | | 88.05 | | |

[3] https://github.com/guochengqian/openpoints

**Local-matching and Global-matching.** The segmentation task is more challenging than the classification task as it needs both global information (shape) and local information (part). Therefore, we propose local and global matching in the segmentation task. To verify the role of each objective, we visualize the point clouds generated by DD3D with local and global matching, respectively. The results are shown below. We can see that global matching cannot learn the shapes of parts, such as the chassis of a laptop (Row 3, Column 2). Local matching cannot capture the spatial relationship between parts, such as the handle and body of a bag (Row 4, Column 4). Therefore, the combination of local and global matching is essential for the distillation of the segmentation task.

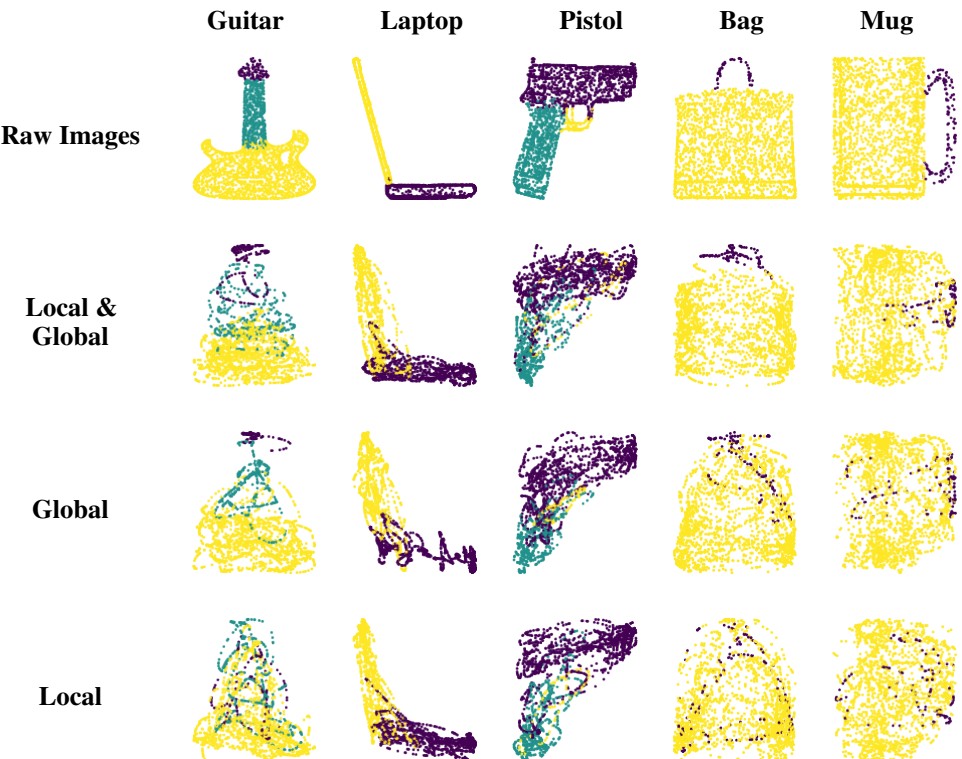

Figure 7: Synthetic images of DD3D with different matching objectives.

**Delving into Aligned and Misaligned Orientations.** To further analyze the influence of rotated and aligned point clouds, we use Grad-CAM to visualize the importance distribution of PointNet trained on these two datasets. We denote the two models as PointNet-aligned and PointNet-rotated. It can be observed that the importance of PointNet-aligned is more concentrated than PointNet-rotated, validating our analysis that rotation-invariant features can preserve the principle components.

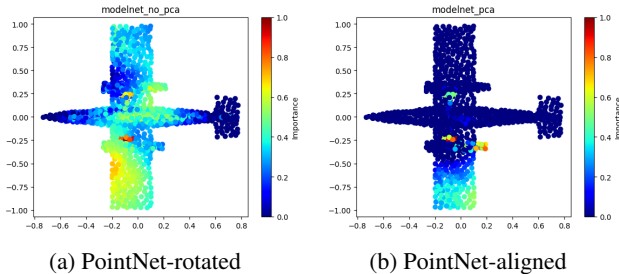

(a) PointNet-rotated       (b) PointNet-aligned

Figure 8: Grad-CAM importance of PointNet trained on aligned and rotated datasets.

**How to Learn Geometric Details?** DD typically focuses on capturing coarse-grained (low-frequency) features that encode the principle information of real datasets. When applied to tasks requiring fine-grained information, we may be concerned about its ability to learn high-frequency details. We visualize DD3D with different periods, *i.e.*, t=10/50/100, in the following. A larger value of $t$ indicates a higher frequency components. We can find that increasing high-frequency input can significantly help DD3D learn geometric details. This inspires us to enhance high-frequency information for fine-grained tasks.

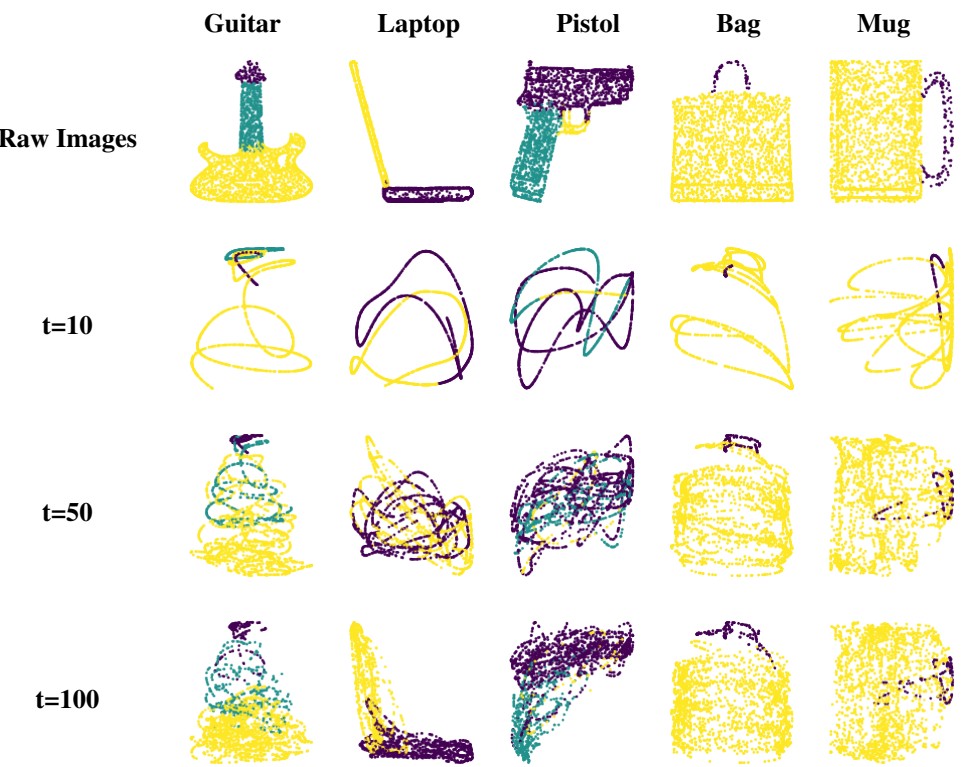

Figure 9: Synthetic images of DD3D with different periods.