# OpenReview forum: "Point Cloud Dataset Distillation"
_ICLR.cc/2025/Conference — Submitted to ICLR 2025_

### Official Review · Reviewer_5cAr · 2024-10-15

**Soundness:** 3
**Presentation:** 3
**Contribution:** 2
**Rating:** 6
**Confidence:** 3

**Summary:**

This paper proposes a dataset distillation method for 3D point clouds. The paper begins by showing that an ideal dataset distillation process should preserve the variance of the underlying dataset and that orientation-misaligned samples introduce undesirable variance perturbations. Based on this observation, the paper uses a rotator to align the point clouds to canonical orientations. Experiments on shape classification and segmentation tasks demonstrate that the proposed DD3D outperforms existing baseline approaches.

Caveat: I am not familiar with dataset distillation. Please interpret my reviews with caution.

**Strengths:**

1. The rotator addresses the sign ambiguity problem associated with PCA. The key idea is to wrap the projected point clouds using a series of sine functions.
2. In addition to the rotation invariancy, DD3D is robust to point cloud resolutions. This is achieved with a point generator that decodes 3D coordinates following a pointwise paradigm. The number of output points is the same as the dimension of the input noise.
3. Ablation studies, particularly Table 5, demonstrate the effectiveness of the proposed design.

**Weaknesses:**

1. While Section 5.6 provides some representative visualizations of the DDD3D-generated synthetic point clouds, it is unclear how they compare with the baseline-generated ones. As such, it is unclear how the visualization should be interpreted. I recommend the authors include side-by-side comparisons of point clouds generated by DD3D and baseline methods. This would allow readers to visually assess the differences and better interpret the visualizations.

2. The proposed method appears to be limited to shape-level tasks. It is not straightforward to apply DD3D to large-scale scene-level tasks. In particular, there may not be a well-defined canonical frame for the diverse scene-level point clouds.

**Questions:**

- Theorems 1 and 2 are built upon several assumptions. Does the actual experiment follow all the assumptions? I understand that the most important assumption is that $f_\theta$ must be rotation-equivalent. The fundamental design of this paper relies on addressing the fact that existing models are not rotation-equivalent. What about the other assumptions? For example, the classifier is assumed to be linear and the classification loss is assumed to be MSE. I do not believe the actual experiment follows these assumptions. It would be great if the authors could provide some clarification on whether the theoretical results can be easily transferred. Rigorous proofs are not necessary but some insightful analysis will be great.

- To follow up on the previous point, perhaps a toy example can be beneficial to illustrate the concept. Consider generating a small collection of 3D shapes and comparing the performance of a shape classification model trained with aligned and misaligned orientations. Can we gain any insights from this toy example?

- Nit: On line 122, $V$ is likely a vector of integers instead of generic real numbers.

---

> ### Author Response · Authors · 2024-11-21
> **Response to Reviewer 5cAr**
>
> Thank you for your thoughtful review and suggestions!
>
> > **W1: Visualization Comparisons.**
>
> A1: In the revised manuscript, **Section 5.6**, we provide visualizations of synthetic datasets generated by DD3D and GM on ModelNet40 and ShapeNet. From these comparisons, we observe the following:
>
> In GM, most points condense into clusters, while some isolated points are left as noise. Moreover, in ShapeNet, the shapes of point clouds are squeezed, making the global shapes difficult to recognize.
> On the other hand, DD3D generates synthetic point clouds with better quality. First, all points are combined into a whole without isolated points. Second, the global shapes are consistent with the real data without squeezing. This may be attributed to the point-wise generator that learns the relationship between points through implicit neural representations.
>
> These visualizations underscore the strengths of DD3D and reveal the limitations of baseline methods. Additional visualizations are provided in **Appendix F** to further validate DD3D’s effectiveness.
>
> > **W2: Applicability to Scene-Level Tasks.**
>
> A2: For indoor scenes, such as S3DIS [1], it is feasible to down-sample the real datasets for distillation and generate dense point clouds for validation. For extremely large scenes, such as Waymo [2], down-sampling will affect the quality of small objects, which is still a challenge for DD3D. We appreciate this suggestion and recognize the importance of extending DD3D for scene-level tasks in future work.
>
> [1] 3D Semantic Parsing of Large-Scale Indoor Spaces. CVPR 2016.
>
> [2] Scalability in Perception for Autonomous Driving: Waymo Open Dataset. CVPR 2020.
>
> > **Q1: Theoretical Assumptions.**
>
> A1: In practice, the classifier is indeed a linear layer $W: \mathbb{R}^{d \rightarrow c}$, but the loss function is cross-entropy (CE) rather than MSE.
> To reduce their gaps, we use KL-divergence to represent CE and MSE:
>
> - CE: $-\sum_i y_i \log x_i = KL(x||y) - C$.
>
> - MSE: $||x-y||^2=KL(x||y), \ \text{s.t.} \ x \sim \mathcal{N}(x, I), y \sim \mathcal{N}(y, I)$.
>
> Their difference is that CE requires a normalized discrete distribution, while MSE needs a continuous distribution sampled from the Gaussian distribution.
> To align CE and MSE, a simple way is to add some Gaussian noise $z \sim \mathcal{N}(0, I)$ to the representation, thus transforming it into a Gaussian distribution.
>
> > **Q2: Toy example on aligned and misaligned orientations.**
>
> A2: Thanks for the great advice. In the revised manuscript, **Appendix F**, we use Grad-CAM [3] to visualize the importance of data for PointNet trained on aligned and misaligned datasets.
> The results show that for models trained on aligned datasets, the importance score is more concentrated, validating our theoretical analysis that rotation-invariant features better preserve principal components. This provides an intuitive understanding of the benefits of alignment.
>
> [3] Grad-CAM: Visual Explanations from Deep Networks via Gradient-based Localization. IJCV, 2019.
>
> > **Q3: Typo**
>
> A3: Thank you for pointing out the typographical issue on line 122. We will correct this in the revised manuscript.

---

> > ### Comment · Reviewer_5cAr · 2024-11-23
> >
> > Dear Authors,
> >
> > Thank you for the enthusiastic reply! I would like to maintain my positive rating and encourage you to engage in discussions with other reviewers who recommended rejection.

---

> > > ### Author Response · Authors · 2024-11-23
> > > **Thanks for Your Comments**
> > >
> > > Dear Reviewer 5cAr,
> > >
> > > Thank you for your encouragement and timely feedback! We really appreciate your thoughtful suggestions, which greatly improve our work.
> > >
> > > Sincerely,
> > >
> > > Authors of Submission4069

---

### Official Review · Reviewer_V3Ng · 2024-11-02

**Soundness:** 3
**Presentation:** 3
**Contribution:** 3
**Rating:** 5
**Confidence:** 3

**Summary:**

This paper introduces DD3D, the first dataset distillation framework specifically designed for 3D point cloud data. The method addresses two fundamental challenges in point cloud distillation: handling different orientations and varying resolutions. The authors make several key contributions. The authors develop a theoretical foundation that demonstrates the importance of matching rotation-invariant features for 3D distillation. And this paper introduces a plug-and-play point cloud rotator that resolves both rotation and sign ambiguity issues. The framework includes a novel rotator to align point clouds to a canonical orientation and employs a point-wise generator to produce point clouds at arbitrary resolutions, significantly advancing the field of dataset distillation for 3D data.

**Strengths:**

1. The paper introduces the dataset distillation framework specifically designed for point cloud data, filling a gap in the field where previous DD methods were primarily focused on structured data like images, videos, and text. This work opens up possibilities for efficient point cloud model training.

2. The authors provide a theoretical analysis demonstrating why matching rotation-invariant features is crucial for 3D point cloud distillation. They formally prove that random rotations weaken the principal components of real data, leading to degraded distillation performance, which strongly motivates the rotator of their approach.

3. The method consistently demonstrates superior performance over traditional dataset distillation methods across multiple benchmarks, validating its effectiveness and practical utility.

**Weaknesses:**

1. The method shows limited performance on fine-grained tasks such as part segmentation compared to shape classification, indicating a potential limitation in capturing detailed geometric features.

2. The computational overhead introduced by the generation process is higher compared to traditional DD methods, which could be a significant limitation for complicated and large-scaled datasets.

3. The significant performance drop compared to using the full dataset, while expected in dataset distillation, still represents a considerable limitation for applications requiring high accuracy. While the paper demonstrates notable improvements over existing dataset distillation methods, there are significant concerns about potential performance degradation when scaling to larger, more complex datasets or more sophisticated tasks.

**Questions:**

1. How does the choice of the noise distribution in the point-wise generator affect the quality of generated point clouds? Have you experimented with different distributions (i.e., Gaussian) beyond uniform?

2. The paper shows that training with low-resolution point clouds can achieve similar performance to high-resolution ones. Could you elaborate on the theoretical reasons behind this result?

3. While the point-wise generator offers resolution flexibility, I have concerns about its inherent limitations in generating complex geometric shapes. How does the performance of the generator architecture affect the observed performance gap in more detailed tasks?

---

> ### Author Response · Authors · 2024-11-21
> **Response to Reviewer V3Ng (1/2)**
>
> We appreciate your detailed review and the recognition of our contributions.
>
> > **W1: Limitation in capturing detailed geometric features.**
>
> A1: DD primarily focuses on coarse-grained (low-frequency) features that encode principal information. Incorporating high-frequency information to model fine-grained details is indeed challenging.
>
> In our revision **Appendix F**, we include visualizations of DD3D with varying frequencies controlled by the hyperparameter $t$. A larger $t$ corresponds to incorporating higher-frequency components. These visualizations demonstrate that increasing the frequency enhances DD3D’s ability to capture fine-grained geometric features.
>
> > **W2: Computational overhead.**
>
> A2: The generator introduces additional computations but also significantly reduces the memory overhead, which is more important for large-scale datasets [1].
>
> To verify this, we first evaluate the time overhead of point cloud generation and gradient matching in ModelNet40. Specifically, we use `line_profiler` to evaluate the time percentages of these two components. The results below indicate that the generation process has minimal impact on computational overhead:
>
> | CPC | Generation (\%) | Matching (\%) |
> | :-- | :--:            | :--:          |
> | 1   | 3.5             | 84.2          |
> | 10  | 2.4             | 86.8          |
> | 50  | 0.9             | 85.9          |
>
> Additionally, we compare the memory usage and accuracy for DD and DD3D across varying CPC values:
>
> | CPC (1/10/50) | Memory (MB) | Performance |
> | :--  | :--:              | :--:         |
> | DD   | 120 / 1200 / 6000 | 53.38 / 72.11 / 75.45 |
> | DD3D | 140 / 230 / 630   | **53.82 / 73.54 / 76.31** |
>
> Notably, under CPC=10, the memory cost of DD is 10x higher than DD3D, but its performance is still not as good as DD3D. In this way, when applied to large-scale datasets, DD3D can handle larger CPC, while others will face out-of-memory issues due to the limited GPU memory. Therefore, DD3D is more suitable for large-scale datasets.
>
> [1] Dataset condensation via efficient synthetic-data parameterization. ICML 2022.
>
> > **W3: Limitation for applications requiring high accuracy.**
>
> A3: We acknowledge the importance of achieving high accuracy for certain applications. Increasing CPC consistently improves performance, as shown in additional experiments conducted on ScanObjectNN and ModelNet40:
>
> | ScanObjectNN | CPC=50 | CPC=100 | Full |
> | :--  | :--:  | :--:  | :--:  |
> | GM   | 57.52 | 62.82 | 66.96 |
> | DD3D | 61.96 | **65.51** | 66.96 |
>
> | ModelNet40 | CPC=50 | CPC=100 | Full |
> | :--  | :--:  | :--:  | :--:  |
> | GM   | 81.74 | 84.17 | 88.05 |
> | DD3D | 83.91 | **86.68** | 88.05 |
>
> These results show that DD3D achieves near-real-dataset performance when CPC is sufficiently large, making it suitable for high-accuracy applications.

---

> > ### Author Response · Authors · 2024-11-21
> > **Response to Reviewer V3Ng (2/2)**
> >
> > > **Q1: Choice of the noise distribution.**
> >
> > A1: The choice of noise distribution impacts performance but not significantly. Experiments with uniform and Gaussian distributions yielded the following results:
> >
> > | CPC=50 | ScanObject | ModelNet40 |
> > | :-- | :-- | :-- |
> > | Uniform  | 43.77 | 76.31 |
> > | Gaussian | 42.82 | 75.36 |
> >
> > > **Q2: Theoretical reasons behind training with low-resolution point clouds.**
> >
> > A2: This observation can be explained by insights from prior work [2], which shows that permutation-invariant features often rely on a few key points due to the max-pooling operator in backbone architectures. As long as these key points are sampled, low-resolution point clouds can achieve similar performance to high-resolution ones.
> >
> > To ensure these key points are encoded in the synthetic dataset, DD3D samples different points in each iteration. This iterative sampling guarantees the inclusion of critical information.
> >
> > [2] Why Discard if You can Recycle? A Recycling Max Pooling Module for 3D Point Cloud Analysis. CVPR, 2022.
> >
> > > **Q3: How does the performance of the generator architecture affect the observed performance gap in more detailed tasks?**
> >
> > A3: Recalling that the point-wise generator of DD3D aims to map a scaler noise into 3d coordinates, formulated as $g: \mathbb{R} \rightarrow \mathbb{R}^{3}$.
> > This unique design connects noise with pre-defined labels, enabling invariance to noise initialization. For example, in segmentation tasks, noise values within specific ranges are assigned part labels (e.g.,  [0, 0.45]  for fuselage).
> >
> > This property is critical for DD3D but is absent in existing point cloud generators:
> >
> > VAE-based methods [3] decode point clouds from the randomly sampled mean and variance vector, which can be formulated as $g: \mathbb{R}^{d, d} \rightarrow \mathbb{R}^{n \times 3}$.
> >
> >  GAN-based methods [4] map a random Gaussian vector into point clouds, which can be formulated as $g: \mathbb{R}^{d} \rightarrow \mathbb{R}^{n \times 3}$.
> >
> > Diffusion-based methods [5] predict the noise of input data, which can be formulated as $g: \mathbb{R}^{n, 3} \rightarrow \mathbb{R}^{n, 3}$.
> >
> > [3] PointFlow: 3D Point Cloud Generation with Continuous Normalizing Flows. ICCV, 2019.
> >
> > [4] Learning Representations and Generative Models for 3d Point Clouds. ICML, 2018.
> >
> > [5] Diffusion Probabilistic Models for 3D Point Cloud

---

> ### Author Response · Authors · 2024-11-27
> **Kind Request for Your Feedback and Further Discussion**
>
> Dear Reviewer V3Ng:
>
> Thank you for your insightful comments, which have been instrumental in refining our work. We greatly value your comments and have addressed your concerns in our revised manuscript.
>
> As the system only allows us to update the revision within the next 24 hours, would you kindly take a moment to review our responses? Your feedback and rating are extremely important to us. We would also appreciate any additional suggestions you might have.
>
> Sincerely,
>
> Authors of Submission4069

---

### Official Review · Reviewer_wYF4 · 2024-11-02

**Soundness:** 2
**Presentation:** 2
**Contribution:** 2
**Rating:** 5
**Confidence:** 4

**Summary:**

The paper introduces DD3D, a method for condensing large 3D point cloud datasets into smaller, efficient sets while maintaining performance. Unlike older methods made for images, DD3D addresses 3D-specific challenges like different orientations and resolutions. It uses a point cloud rotator to align data consistently and a point-wise generator to create flexible, high-resolution synthetic point clouds. Experiments show DD3D is effective, scalable, and memory-efficient, making it suitable for tasks like shape classification and part segmentation.

**Strengths:**

1） The paper backs its approach with solid mathematical explanations, showing why the proposed techniques work.
2）The authors test DD3D thoroughly across different datasets and scenarios, showing that it consistently outperforms traditional methods.

**Weaknesses:**

1. The authors only verify the effectiveness of classification tasks on very small datasets.
Therefore, my biggest concern is the generalization of the proposed method for detection tasks on large dataset(nuscenes or waymo).
2. Why adopt data distillation? Data distillation will cause some information loss compared with raw data.
3. While the paper compares DD3D to several distillation methods, it could strengthen its claims by including more comparisons with state-of-the-art rotation-invariant models.
4. Although the paper mentions some drawbacks, such as convergence challenges when not initialized with real data, a deeper exploration of these limitations and potential mitigation strategies would add more balance to the work.

**Questions:**

See the weaknesses.

---

> ### Author Response · Authors · 2024-11-21
> **Response to Reviewer wYF4**
>
> We are grateful for your constructive advice and the opportunity to address your concerns.
>
> > **W1: Generalization of the proposed method for detection tasks.**
>
> A1: We recognize that adapting DD3D for the detection task poses unique challenges due to the continuous nature of the label space (e.g., bounding box locations) compared to the discrete labels in classification and segmentation tasks. Given these constraints, pre-defining labels for synthetic datasets in the detection task is non-trivial. As such, we believe the detection task is beyond the current scope of this paper. We plan to explore methods for addressing this challenge in future work.
>
> In addition, we have conducted classification experiments on MVPNet, which is a much larger dataset with 87k samples. The results validate the effectiveness of DD3D over baselines in large-scale datasets.
>
> > **W2: Why adopt data distillation?**
>
> A2: DD significantly reduces dataset redundancy and accelerates neural network training by creating compact, task-specific synthetic datasets. While some information loss is inevitable, DD can achieve performance close to real datasets by increasing the synthetic dataset budget (e.g., CPC=100). The following results highlight this capability:
>
> | ScanObjectNN | CPC=50 | CPC=100 | Full |
> | :--  | :--:  | :--:  | :--:  |
> | GM   | 57.52 | 62.82 | 66.96 |
> | DD3D | 61.96 | **65.51** | 66.96 |
>
> | ModelNet40 | CPC=50 | CPC=100 | Full |
> | :--  | :--:  | :--:  | :--:  |
> | GM   | 81.74 | 84.17 | 88.05 |
> | DD3D | 83.91 | **86.68** | 88.05 |
>
> Furthermore, DD has proven useful in areas like continual learning [1] and privacy preservation [2], demonstrating its broader utility.
>
> [1] An Efficient Dataset Condensation Plugin and Its Application to Continual Learning. NeurIPS 2023.
>
> [2] Privacy for Free: How does Dataset Condensation Help Privacy? ICML 2022.
>
>
> > **W3: More comparisons with state-of-the-art rotation-invariant models.**
>
> A3: We have discussed some rotation-invariant methods in Lines 197-201. While SOTA rotation-invariant models outperform DD3D on rotated datasets, our goal is different: enabling rotation-variant models (e.g., PointNet) to distill datasets with diverse orientations. Rotation-invariant models rely on specialized architectures, which are challenging to transfer to rotation-variant frameworks. In contrast, DD3D’s rotator is a general, modular approach that can be implemented as a wrapper for any model, offering greater flexibility.
>
> > **W4: A deeper exploration of these limitations and potential mitigation strategies.**
>
> A4: We have revised the limitations section to provide a more thorough discussion of challenges and possible solutions:
>
> - Application to Continuous Label Tasks: Existing methods are not directly applicable to tasks with continuous labels (e.g., detection). A potential solution is discretizing continuous labels, such as treating points within bounding boxes as a separate class.
>
> - Encoding Fine-Grained Details: Current methods focus primarily on coarse-grained information (e.g., shapes), resulting in limitations for fine-grained tasks. Enhancing high-frequency information in gradient matching could address this limitation. We further explore this idea in **Appendix F.**

---

> ### Author Response · Authors · 2024-11-27
> **Kind Request for Your Feedback and Further Discussion**
>
> Dear Reviewer wYF4:
>
> Thank you for your thoughtful review. Based on your feedback, we have made several improvements in the revision. If there are any remaining concerns or areas where further clarification would be helpful, we would be happy to address them promptly. We deeply value your input and look forward to continuing the discussion.
>
> Best regards,
>
> Authors of Submission4069

---

### Official Review · Reviewer_51ao · 2024-11-04

**Soundness:** 3
**Presentation:** 3
**Contribution:** 2
**Rating:** 5
**Confidence:** 2

**Summary:**

In point cloud dataset distillation, this paper addresses the challenge of diverse orientations and resolutions of point clounds.
Specifically, this paper proposes the first 3D distillation framework , which can be trained with low-resolution point clouds and generates high-resolution data for evaluation.
It also validate the effectiveness of matching the rotation-invariant features in preserving the principal components of real data.

**Strengths:**

1. This paper is the first to propose dataset distillation framework in point cloud data.
2. The paper is well written and fully describes the research area and proposed methods.

**Weaknesses:**

1. The proposed method requires different distillation algorithms for different tasks, so it may be troublesome to use in practice.
2. The table 2，the improvement is limited compared with coreset-based methods,  and far behind the full dataset. I have concerns about the practicality of the proposed method.
3. The figure 7 and figure 9 demonstrate that the synthetic point clouds have poor quality.

**Questions:**

It seem that there is another method  (https://github.com/kghandour/dd3d/blob/main/assets/Report.pdf) in dataset distillation of point clouds using GM,
Why is there no mention of this method?

---

> ### Author Response · Authors · 2024-11-21
> **Response to Reviewer 51ao**
>
> We sincerely appreciate your thoughtful feedback and insightful questions.
>
> > **W1: The proposed method requires different distillation algorithms for different tasks.**
>
> A1: We acknowledge your concern regarding the need for task-specific distillation algorithms. We would like to clarify that DD3D is a general method for various point cloud tasks. While it requires task-specific gradients during distillation, the overall algorithm remains unchanged.
>
> For instance, DD3D uses the same distillation pipeline in both classification and segmentation tasks. However, the segmentation task is inherently more complex as it requires both global and local information. To address this, we combine gradients from part-level (local) and shape-level (global) supervision to improve performance.
>
> To validate this approach, we conducted an ablation study to assess the impact of local, global, and combined supervision in segmentation tasks. The results are summarized below:
>
> |      |  Local | Global | Local+Global |
> | :--  | :--          | :--   | :--    |
> | DD3D | 48.68 | 47.17  |  **50.99**        |
>
> Additionally, we include visualizations of point clouds generated by DD3D with different objectives in the revised manuscript, **Appendix F**. These visualizations highlight that global matching alone fails to preserve part-level details, while local matching struggles with spatial relationships between parts. Only the combination of local and global objectives generates high-quality point clouds.
>
> > **W2: Concerns about the practicality of the proposed method.**
>
> A2: When the budget of synthetic datasets is limited, e.g., CPC=1, distillation-based methods significantly outperform the coreset-based methods. Moreover, on large-scale datasets such as MVPNet, DD3D demonstrates an average improvement of 15% over coreset methods, showcasing its practical utility.
>
> Moreover, increasing CPC narrows the performance gap between synthetic and real datasets. We conducted additional experiments with CPC=100 on ScanObjectNN and ModelNet40, as shown below:
>
> | ScanObjectNN | CPC=50 | CPC=100 | Full |
> | :--  | :--:  | :--:  | :--:  |
> | GM   | 57.52 | 62.82 | 66.96 |
> | DD3D | 61.96 | **65.51** | 66.96 |
>
> | ModelNet40 | CPC=50 | CPC=100 | Full |
> | :--  | :--:  | :--:  | :--:  |
> | GM   | 81.74 | 84.17 | 88.05 |
> | DD3D | 83.91 | **86.68** | 88.05 |
>
>
> > **W3: Quality of the synthetic point clouds.**
>
> A3: We agree that the visual quality of synthetic point clouds is an important aspect. It is important to differentiate the goals of DD and traditional generation tasks:
>
> Traditional generation task aims to approximate the real data distribution, i.e., $p(X_{\mathcal{T}}) \approx p(X_{\mathcal{S}})$
> In contrast, DD focuses on creating compact, task-specific synthetic datasets, where $p(X_{\mathcal{T}}) \neq p(X_{\mathcal{S}})$, prioritizing informativeness over realism.
>
> This distinction explains why synthetic data from DD often exhibits deformations. The same phenomenon can be observed in the distillation of image datasets. Despite these deformations, DD3D preserves the critical global shapes and avoids isolated nodes, as demonstrated in **Section 5.6** of the revised manuscript.
>
> > **Q1: Related work.**
>
> A1: We appreciate the reference to the related GM-based method. We have included a discussion of this work in the revised manuscript to provide a more comprehensive review.

---

> > ### Comment · Reviewer_51ao · 2024-11-24
> >
> > I agree that your method focuses on specific tasks, such as point cloud classification and segmentation, where the poor visual quality of synthetic point clouds may not pose a significant issue.
> >  Additionally, as an early attempt in point cloud dataset distillation, the improvement compared with coreset-based methods is acceptable, even though there remains a gap compared to training with raw data.
> > But I still have doubts about the practicality of the method because of the task-specific distillation algorithms.
> > Although global and local information can improve the performance, I regard it would be more convenient to use a unified distillation algorithm for different tasks in practice.
> > If I want conduct the experiments of point cloud object detection, is your algorithm still applicable?

---

> > > ### Author Response · Authors · 2024-11-24
> > > **Response to Reviewer 51ao**
> > >
> > > Thank you for the prompt reply and recognition of our contribution!
> > >
> > > The principle of gradient matching is to match gradients **by categories**, thereby significantly simplifying the distillation process. Therefore, our algorithm can unify tasks with **discrete labels** where shape classification is relatively straightforward, as each example is associated with a single category. In contrast, segmentation tasks are more complex, as each example may have multiple labels. Notably, prior 2D image DD methods only focus on the classification task, and we make the first attempt to address the segmentation task.
> > >
> > > The detection task, however, is inherently more challenging as it involves both discrete and continuous labels.
> > > Our algorithm is well-suited for segmenting foreground and background points (Stage-1 in PointRCNN [1]) but is less applicable to 3D bounding box localization (Stage-2 in PointRCNN), which requires continuous labels. In this case, it is not feasible to predefine the labels of the synthetic dataset or match gradients by categories.
> > >
> > > In the limitations part of the revision, we elaborate on the challenges posed by discrete and continuous labels. We hope this addresses your concerns about the practicality of our approach.
> > >
> > > [1] PointRCNN: 3D Object Proposal Generation and Detection from Point Cloud. CVPR, 2019.

---

> > > ### Author Response · Authors · 2024-11-27
> > > **Kind Request for Your Feedback and Further Discussion**
> > >
> > > Dear Reviewer 51ao:
> > >
> > > Thank you once again for your helpful comments. As the system only allows us to update the revision within the next 24 hours, we kindly ask if you have any additional questions that require clarification. We would be delighted to address them.
> > >
> > > Looking forward to hearing your thoughts.
> > >
> > > Authors of Submission4069

---

### Author Response · Authors · 2024-11-21
**Global Response to All Reviewers**

We sincerely thank all reviewers for their thoughtful suggestions and recognition of our contributions. We have addressed most of the questions and suggestions in the revised paper (highlighted in blue color). Below, we summarize the main revisions.

- More experimental results. We make more ablations to verify the effectiveness of DD3D, and conduct experiments on CPC=100 to further narrow the performance gap with real datasets.

- More visualizations. We visualize the synthetic dataset generated by DD3D and GM and place them side-by-side for an intuitive comparison and demonstrate our strengths.

- Broader discussions. We revise the limitation section to discuss the weaknesses of DD3D and the challenges of distilling 3d data.

---

### Meta-Review · Area_Chair_CmKb · 2024-12-24

**Metareview:**

This paper introduces 3D dataset distillation to replace large-scale real datasets with small and synthetic datasets. The key idea includes a proposed point cloud orientation predictor for rotation-invariant features and a point generator for various scales. The proposed approach enables training with low-resolution point clouds while making evaluations with high-resolution point clouds, which leads to reduced memory requirements.

The strength of the paper is summarized as follows:
- First attempt to distill point cloud data
- Solid math explanations
- Validation of the various datasets and scenarios
- Easy to read

The weakness of the paper is summarized as follows:
- Distillation algorithm differs per the task
- Practicality of the proposed approach due to several technical reasons
- Visual quality of the synthetic point clouds

AC especially sees that the task-specific distillation approach and the visual quality of the synthetic datasets did not fully convince the reviewers. This also related to the questions on the practicality of the proposed approach for other scene-level tasks since the orientation agnostic module is more suited for object-centric point clouds. Therefore, by following the reviewers' overall concerns, AC notes that the weakness outweighs the strength of the paper.

**Additional Comments On Reviewer Discussion:**

Overall, this paper received diverged scores. Unfortunately, reviewers wYF4, 51ao, and 5cAr were inactive during the discussion phase, even though AC and the authors tried to promote the discussion several times. Threfore, AC carefully followed the comments and feedback between reviewers and authors during the review phase.

Specifically, reviewer 51ao provides a short review, and the reviewer asks questions about task-specific distillation algorithms and concerns about the practicality of the proposed approach. The reviewer mentioned that the authors' feedback did not resolve the concerns about the practicality of the proposed approach. The reviewer wYF4 also provides a brief review stating the generalization on the other tasks (detection), the justification of the data distillation on the point clouds, more comparisons on the rotation-invariant models, and clarifying limitations. However, the reviewer, wYF4, did not reply to the authors' feedback. The reviewer, V3Ng, provided a detailed review and questioned the limited detail captured from geometric features, the generator's computation overhead, and the possible limitations when achieving high accuracy. In addition, the main concern raised by reviewer V3Ng is the theoretical foundations behind the proposed idea. Although the authors provided detailed feedback, the reviewer V3Ng states that the reviewer remains the initial score of 5. The reviewer 5cAr provided a constructive review and asked questions about visualizations, applicability to scene-level tasks, theoretical assumptions, and orientation toy experiments. The authors provided solid feedback and revision, but the reviewer did not respond.

---

### Decision · Program_Chairs · 2025-01-22

Reject